# Image Classification-Based Defect Detection of Railway Tracks Using Fiber Bragg Grating Ultrasonic Sensors

**Da-Zhi Dang** [1,2], **Chun-Cheung Lai** [1,2], **Yi-Qing Ni** [1,2,3,*], **Qi Zhao** [1,2], **Boyang Su** [1] and **Qi-Fan Zhou** [1,2]

1   Department of Civil and Environmental Engineering, The Hong Kong Polytechnic University, Hung Hom, Kowloon, Hong Kong SAR, China
2   Hong Kong Branch of National Transit Electrification and Automation Engineering Technology Research Center, The Hong Kong Polytechnic University, Hung Hom, Kowloon, Hong Kong SAR, China
3   The Hong Kong Polytechnic University Shenzhen Research Institute, Shenzhen 518057, China
*   Correspondence: ceyqni@polyu.edu.hk

**Abstract:** Structural health monitoring (SHM) is vital to the maintenance of civil infrastructures. For rail transit systems, early defect detection of rail tracks can effectively prevent the occurrence of severe accidents like derailment. Non-destructive testing (NDT) has been implemented in railway online and offline monitoring systems using state-of-the-art sensing technologies. Data-driven methodologies, especially machine learning, have contributed significantly to modern NDT approaches. In this paper, an efficient and robust image classification model is proposed to achieve railway status identification using ultrasonic guided waves (UGWs). Experimental studies are conducted using a hybrid sensing system consisting of a lead–zirconate–titanate (PZT) actuator and fiber Bragg grating (FBG) sensors. Comparative studies have been firstly carried out to evaluate the performance of the UGW signals obtained by FBG sensors and high-resolution acoustic emission (AE) sensors. Three different rail web conditions are considered in this research, where the rail is: (1) intact without any defect; (2) damaged with an artificial crack; and (3) damaged with a bump on the surface made of blu-tack adhesives. The signals acquired by FBG sensors and AE sensors are compared in time and frequency domains. Then the research focuses on damage detection using a convolutional neural network (CNN) with the input of RGB spectrum images of the UGW signals acquired by FBG sensors, which are calculated using Short-time Fourier Transform (STFT). The proposed image classifier achieves high accuracy in predicting each railway condition. The visualization of the classifier indicates the high efficiency of the proposed paradigm, revealing the potential of the method to be applied to mass railway monitoring systems in the future.

**Keywords:** rail defect detection; image classification; ultrasonic guided wave; fiber Bragg grating; convolutional neural network; Short-time Fourier Transform

## 1. Introduction

Rail transit is considered to be unreplaceable in the modern world, supplying the most basic daily public travelling service and long-distance transportation due to its high efficiency and large passenger capacity. The efficiency of such significant means of transport, however, is always compromised by substandard or impaired infrastructures. The railway tracks are commonly seen with defects such as cracks [1] and weld defects [2]. Figure 1 shows rail cracks discovered near Fo Tan Station and Hung Hom Station on the East Rail Line, Hong Kong in 2017 and 2019, respectively. It has been reported that the most common accidents in the railway system, ranging from vibration noises to major derailments, were the consequences of rail track damage according to the Legislative Council Panel on Transport of Hong Kong SAR [3,4].

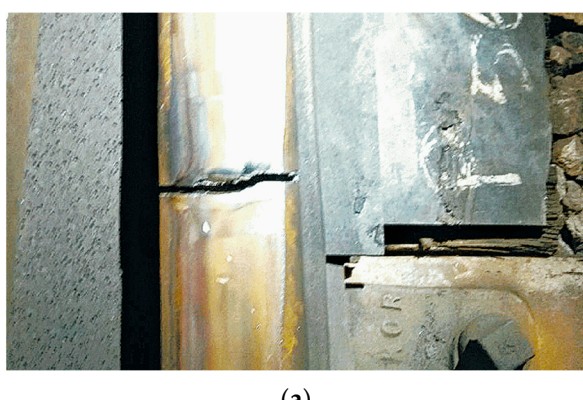

(**a**)

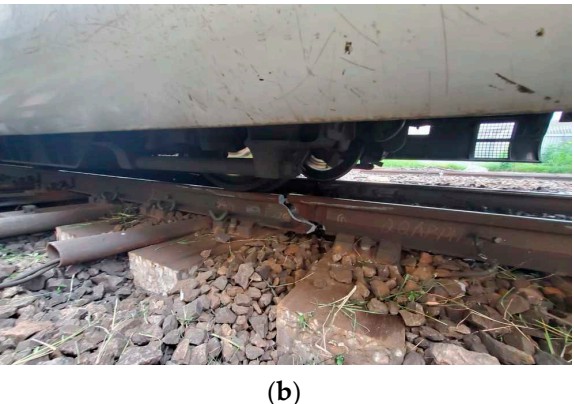

(**b**)

**Figure 1.** On-site photos taken on the East Rail Line: (**a**) crack on the rail near Fo Tan Station in 2017; (**b**) crack on the rail near Hung Hom Station in 2019.

Therefore, an early-in-time and on-site diagnosis should be implemented to monitor the safety status of rails. With more attention paid to the maintenance of the urban and intercity railway systems, inspection vehicles and manual inspection have been widely utilized to detect early-stage defects. These approaches are capable of reporting thorough details of the rail condition, but they can either be labor costly or lack time efficiency for long-range monitoring projects. Therefore, non-destructive testing (NDT) has been applied, using ultrasonic detecting devices [5], optical fiber sensors [6–8], piezoelectric sensors [9–11] and even onsite cameras [12,13] to conduct online monitoring. Such sensing technologies have the distinguished advantage that they can acquire data while not interrupting the operation of the railway systems. For instance, researchers have studied image- and video-based inspection methods using digital cameras installed on the operating trains or fixed on-site aided with computer vision (CV) [12], yet the limited hardware performance prevents these methods from being widely utilized to detect minor or inner defects on the rail tracks. On the other hand, ultrasonic bulk wave-based devices can inspect most abnormalities within the sensing range and have also been utilized on the rail track routine inspections [14]. However, ultrasound actuators and transducers are required to be installed on the tracks, which inevitably brings inconvenience considering that such a monitoring approach can be time-consuming and even insecure in some cases. Similarly, acoustic emission (AE) technology, despite its high resolution in receiving ultrasound [15–17], has the same limitation for the deployment in damage detection of rail segments. Moreover, it is generally assumed that AE sensors are mostly applicable when ultrasonic waves are generated due to the crack growth within the object, which makes it extremely difficult for engineers to design a long-range monitoring system based on such technologies.

Ultrasonic guided waves (UGWs) are especially suitable for railway systems [18–20]. UGWs can transmit for a long distance in the longitudinal direction of the rail web. Once the UGW signals are obtained, further analysis of the wave reflection, mode conversion, and energy loss during the transmission can be conducted accordingly [1,9,18], serving as the damage indices to determine whether defects exist. For example, Sun et al. [1] analyzed the energy attenuation and waveform distortion resulting from the existence of surface cracks, indicating the feasibility and prospects of the mass application of guided wave-based defect detection methods.

However, the performance of UGW monitoring is closely related to the utilization of different types of sensors. AE sensors are generally sensitive to ultrasonic waves, and thus can be utilized as precise transducers. Most AE sensors are lead–zirconate–titanate (PZT) transducers, and the working principle for such sensors is basically to transform the deformation into electric signals via piezoelectric materials. The main drawbacks of these piezoelectric AE sensors are that they are not yet verified to work efficiently on railway sites where strong electromagnetic interference (EMI) should be considered. Furthermore, they require stabilizing electricity power supplies to be functional on-site, let alone the

cost of AE sensors is very likely to be unacceptable for mass transit systems. Last, long-distance electric wires can be fragile and redundant in complex rail systems. Because of the above reasons, optical fiber-based sensing technologies have been proposed to replace the traditional PZTs as transducers to be installed on rail tracks [6–8,21–23]. The non-electric working principle ensures that the sensors can be immune to EMI. Moreover, the optical fiber can last for several kilometers, enabling long-distance monitoring. The cost for bare fibers is considerably lower than most electronic transducers. The high-temperature resistance and the multiplexing feature of optical fibers make them the better choice for on-site installation. Although optical fiber-based sensors are suitable for rail monitoring, it is still challenging to demodulate the optical signals caused by ultrasonic vibrations. The normal dynamic interrogators can achieve the sampling rate of 2 kHz, which is far from adequate considering the dominant frequency of most ultrasonic waves in solid materials exceeds 20 kHz. To detect minor defects UGWs with shorter wavelengths are required, meaning the actuated frequency may exceed 200 kHz. Therefore, fiber Bragg grating (FBG) sensors have been studied and utilized to monitor ultrasonic signals using a laser source to demodulate the optical spectrum. An FBG would only reflect the light with a specific wavelength (referred to as the Bragg wavelength) so that the change in that wavelength would indicate the effect of the external force or temperature. The specific methodology of using FBG to receive UGWs will be introduced in Section 2. As for current related research, Lamb wave-based detection and source location were studied by Betz et al. [24], followed by Wee et al. [21] who investigated the directional sensitivity of FBGs to receive Lamb waves through experimental studies. An improved adhesive was proposed by Yu et al. [25] by implementing laser ultrasonic visualization. Tian et al. [26] utilized adaptive phased array imaging approach to detect damage using FBGs. Those researchers have mostly conducted UGW inspections in laboratories based on Lamb or Rayleigh waves. However, rail tracks are generally more complex, thus it can be extremely difficult to apply commonly used indices to find defects through UGW signals.

In previous studies, Sun et al. [1] have proved the feasibility of using FBGs to detect cracks of different diameters on the rail tracks, by comparing the energy attenuation of the guided waves signals. In this paper, an image classification approach is proposed, where the feature extraction process is simplified using a deep neural network (DNN). In recent years, deep learning has been introduced to structural health monitoring (SHM) [27].

Sloun et al. [28] gave an overview of the use of data-driven deep learning strategies in ultrasound systems, ranging from the front-end to advanced applications. Medak et al. [29] proposed a DNN for sequence analysis, which greatly improved the data acquisition efficiency on structural NDT using ultrasonic B-scan. It has been proven that DNNs can improve the efficiency and accuracy of the NDT using ultrasonic devices. Moreover, DNNs specially designed for UGW testing were proposed, which further outperform widely used conventional analysis methods [30–32]. These applications are all based on developing innovative networks to process the ultrasonic datasets. The advantage of DNNs is that they can automatically extract features from the data collected to help identify the structural status. Instead of calculating damage indices defined by conventional statistical and analytical formulas, a robust DNN can achieve efficient classification through convolution and pooling layers, significantly simplifying the processing procedures while preventing subjective judgments [33]. Hence, deep learning has been utilized to assess the railway safety conditions. For example, Chen et al. [34] utilized a pre-trained DNN to evaluate the condition of rail wheels with the input of AE data, which proved the feasibility of fusing vibration signals with audio signals for detecting damage conditions. Similar research has also been conducted, mostly using the time-series signals as the network's input. However, the influence of damage on UGW propagation will not only be observed in the time domain but also in the frequency domain. In other words, spectrograms of the UGW signals also have the potential for crack detections. In fact, such time-frequency feature representation schemes have been implemented in various research areas, such as DL-based speech processing [35], denoising for non-stationary signals [36] and emotion recognition [37].

Ultrasonic signals share homologous similarities with other voice-based datasets. Therefore, it is assumed applicable to transfer this methodology and deploy on the signal processing of UGWs. Zhang et al. [38] utilized spectrums of the ultrasound signals to generate damage sensitive features to conduct fault identification. It has also been proven that multiple frequency domain features, acquired by deploying continuous wavelet transform and Hilbert transform, can contribute to the DNN-based ultrasonic inspection [39].

In this paper, we make further progress based on previous research by introducing an image classification approach for feature extraction and identification of rail conditions based on UGW signals. The methodologies are firstly introduced in Section 2, where the basic physical principles of the FBG-based guided wave inspections are introduced. Then, the experimental studies are demonstrated in Section 3, including the experimental setup of the hybrid system, the rail track conditions, and the comparison of the signals obtained by both FBGs and AE sensors is also demonstrated. In Section 4, an image classifier is proposed to detect damage that exists along the propagation route of UGWs. A robust convolution neural network (CNN) is specially designed with the input being the RGB images of spectrograms of the UGW signals obtained through Short-time Fourier transform (STFT). The visualization of the extracted features is also presented in this section to show the effectiveness of the proposed method. This research work will contribute to a deeper comprehension about the UGW-based non-destructive testing methods, providing a novel perspective on using computer vision to process ultrasound signals.

## 2. Methodology

### 2.1. Working Principle of FBGs

An FBG sensor is a typical optical sensor that has the grating with periodic changes, resulting in only light with a specific wavelength $\lambda_B$ being reflected. This wavelength is defined as the Bragg wavelength which can be expressed as:

$$\lambda_B = 2n\Lambda \tag{1}$$

where $n$ represents the effective refractive index of the optical fiber and $\Lambda$ refers to the grating period. When the broadband light source enters the grating section the light with the Bragg wavelength cannot pass through, and thus is reflected. Consequently, an FBG has a high selectivity on wavelengths. When the external force or temperature changes, $n$ and $\Lambda$ would react to the changes linearly.

Based on that principle, it is possible to demodulate micro-vibrations induced by the ultrasonic waves from FBG measurements. However, the wavelength shift can be extremely minor, the frequency of which also requires a piece of high-speed demodulation equipment. In this case, it is rational to utilize the edge filter demodulation technique [40], where a narrow-bandwidth light enters the optical fiber and meets the 3-dB points of the reflective wavelength, as demonstrated in Figure 2. The quality of the receiving ultrasound signals using this approach has been proven by researchers [41,42]. A balanced photodetector is required to receive the light from the transmission light which passes through the grating of the FBG and the incident light. The photodetector not only transforms the change in the optical signals to digitalized electrical signals, but also reduces the noise. The output voltage of the balanced photodetector can be expressed by:

$$V = 2\Delta\lambda_B \cdot R_D \cdot G \cdot P \cdot g \tag{2}$$

where $G$ represents the grating slope; $g$ is the gain factor; $P$ and $R_D$ are the laser power and the response of the photodetector, respectively.

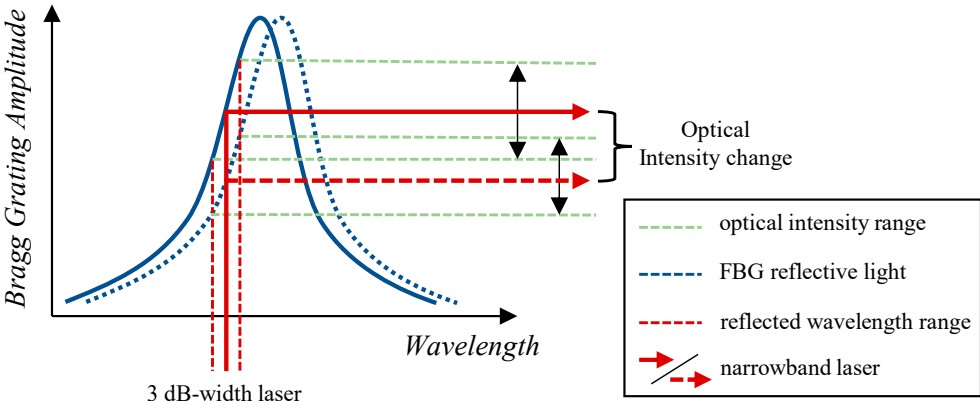

**Figure 2.** Schematic of ultrasound high-speed FBG demodulation.

### 2.2. Bulk Wave and Guided Wave Inspection Methods

Ultrasound-based non-destructive inspection approaches can be briefly divided into two categories according to the wave types: bulk wave inspection and guided wave inspection. The main difference between the two methods is the wave coverage area. Generally, a traditional ultrasonic bulk wave evaluation with normal-beam excitation can only be over a limited volume based on the specification of the ultrasonic probe (Figure 3a). The operator shall move the ultrasonic device in the longitudinal direction to detect the damage (e.g., cracks). However, this approach only applies to ordinary objects with relatively smaller volumes. For railway damage inspection, which requires long monitoring distance, this approach would be time-consuming. Therefore, UGWs monitoring has overcome this drawback. The guided waves are formed through reflections and refractions inside the rail segment, forming stabilizing wave packets propagating along the rail longitude (Figure 3b). The guided wave inspection requires the actuator and the receiver to be located at the beginning and the end of the testing area, respectively. The inspection is completed by analyzing the waveforms. This approach has shown a higher sensitivity level to guided wave inspection than other normally used NDT methods, according to the literature [1,43].

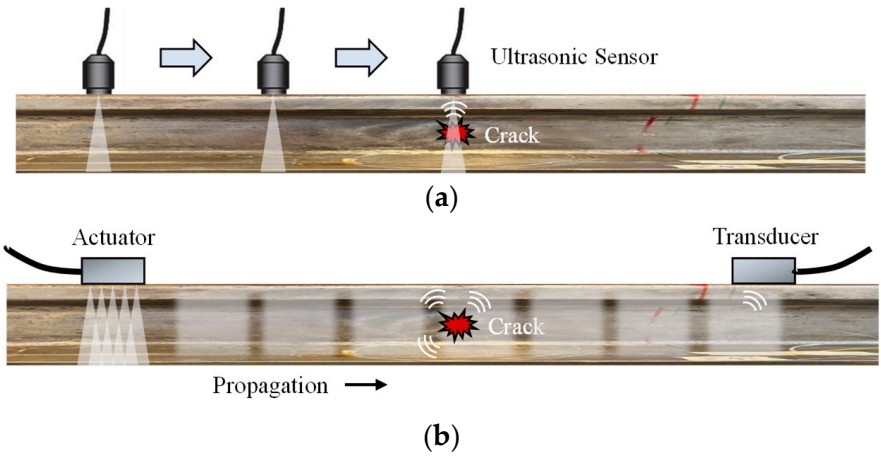

**Figure 3.** Comparison of: (**a**) bulk wave inspection; versus (**b**) guided wave inspection.

The propagation of UGWs in rails depends on the frequency and the wavelength of the ultrasound. Taking the rail web as an example, when an actuator is placed on the rail web, it is expected that the wavelength of the guided waves would generally be greater than the structural thickness. Rose [43] made a comparison of the currently used ultrasonic bulk wave technique and the UGW inspection procedure for plate and pipe inspection, as shown in Table 1.

**Table 1.** Comparison between bulk wave inspection and guided wave inspection [43].

| Features | Bulk Wave Inspection | Guided Wave Inspection |
|---|---|---|
| Efficiency | Laborious and time-consuming | Fast and convenient |
| Accuracy | Point-by-point scan (accurate rectangular grid scan) | Global in nature (approximate line scan) |
| Reliability | Unreliable (can miss points) | Reliable (volumetric coverage) |
| Complexity | High-level training required for inspection | Minimal training |
| Distance | Fixed distance from reflector required | Any reasonable distance from reflector acceptable |
| Identification | The reflector must be accessible and seen | The reflector can be hidden |

Compared to the bulk waves, the guided waves travel as a packet with ultrasonic waves of different velocities. However, the UGW has a general solution to describe the propagation velocity of a wave group where each wave has a similar frequency. The equation given below is utilized to define the group velocity of a UGW:

$$c_g = c_p + k\frac{dc_p}{dk} \tag{3}$$

where $c_g$ and $c_p$ are the group and phase velocities and the parameter $k$ is the wave number. Thus, it is discovered that the individual waves travel at various speeds of $c_p$, while the superimposed packet propagates at the speed of $c_g$. Considering the dispersion of waves, $c_p$ varies with different wave frequency $f$. Then the $c_g - c_p$ relation can be obtained as a reference for conducting guided wave inspections to estimate the propagation in different materials.

## 3. Comparative Study

Based on the methodologies mentioned in the previous section, comparative experiments are carried out to study the propagation of UGWs under the influence of rail damage. To excite stable ultrasound stress waves, a PZT actuator has been pre-installed on the rail web. Both FBG sensors and high-resolution AE sensors (PICO Acoustic Emission Transducers, produced by Physical Acoustics, Princeton, NJ, USA) have been utilized to receive the UGWs, and comparisons have been made to analyze the detailed performance of both sensors.

### 3.1. Experiment Introduction

3.1.1. The PZT/FBG Hybrid Sensing System

The complete experimental setup is shown in Figure 4. A five-cycle sinusoidal tone of 250 kHz burst modulated by the Hanning window is generated (Figure 5) by the arbitrary waveform generator module of the computer-based central terminal instrument (PXI-5412, produced by National Instruments, Austin, TX, USA), which outputs digitalized signals to the power amplifier (HVA-400-A, produced by Ciprian, Grenoble, France). The signals are then amplified 200 times and the maximum voltage reaches 200 V. The amplified signals are sent to a PZT disc (diameter: 8 mm, thickness: 0.5 mm) to generate ultrasonic waves. The PZT is attached to the rail surface with adhesives.

The above descriptions have concluded the actuation part of the system. As for the transducing and sensing part of the system, the main instruments involve the balanced photodetector, the FBG sensor, and the laser source (TLB-6700, produced by Newport, Irvine, CA, USA) and its controller. The laser controller ensures that the narrow-bandwidth light transmitted from the source is of the fixed wavelength and fixed emitting power. In this study, the laser wavelength is set to 1549.95 nm (3 dB position on the right-hand side of the FBG spectrum) and the output current is 40 mA. The laser is evenly split via the 50%/50% coupler (F-CPL, produced by Newport, Irvine, CA, USA), with half transmitting to the FBG and the other half to the balanced photodetector (2117-FC, produced by Newport, Irvine, CA, USA). The grating length of the FBG is 10 mm, which is approximately the wavelength of the UGWs, to obtain as many faultless waveforms as possible. Lastly, the received optical signals are converted into digital signals by the balanced photodetector and are then sent back to the data acquisition and oscilloscope modules of the computer-based instrument. The sampling frequency is 10 MHz in this experiment and the duration of each test is 1 ms.

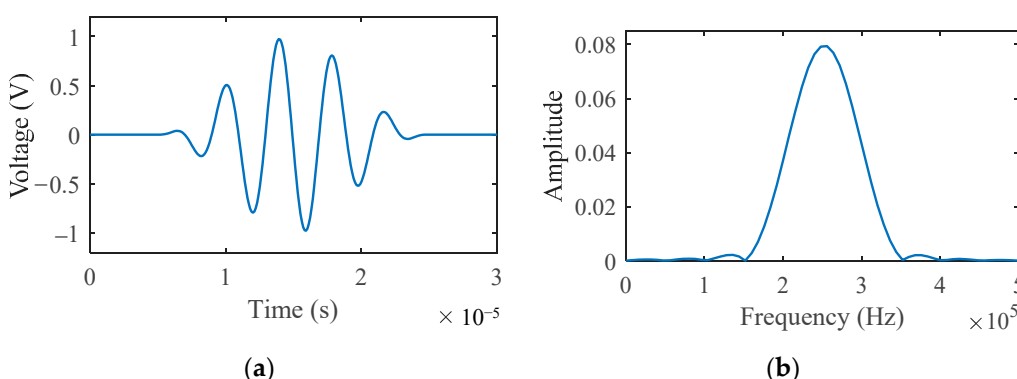

**Figure 4.** The experimental setup: (**a**) the schematic; (**b**) the laboratory setup.

**Figure 5.** The excitation signals in (**a**) time domain; and (**b**) frequency domain.

The whole procedure, including the excitation, laser emission, and data acquisition, is controlled by preprogrammed software based on LabVIEW, which is installed on the computer-based instrument. This program guarantees that the excitation signals and the laser power remain the same in separate tests.

### 3.1.2. Experimental Procedures

A total of three rail track conditions are considered in this experimental study, namely 'Intact', 'Cracked', and 'Bump' (Figure 6). The 'Intact' condition refers to the baseline testing with no extra damage placed along the propagation route of UGW; then, an artificial surface crack with a depth of 5 mm and a length of 20 mm is placed between the FBG and the PZT for the 'Cracked' condition; lastly, to diversify the damage types in this experiment, a 'Bump' condition, which is created by using blu-tack solid adhesive attached to the rail web surface, is also considered. The volume of the blu-tack adhesive is approximately 1 cm$^3$, which is expected to absorb a certain amount of UGWs during propagation [35,36]. It is assumed in this paper that the attached adhesives would have similar effects on the UGW propagation as the weld defect on the rails.

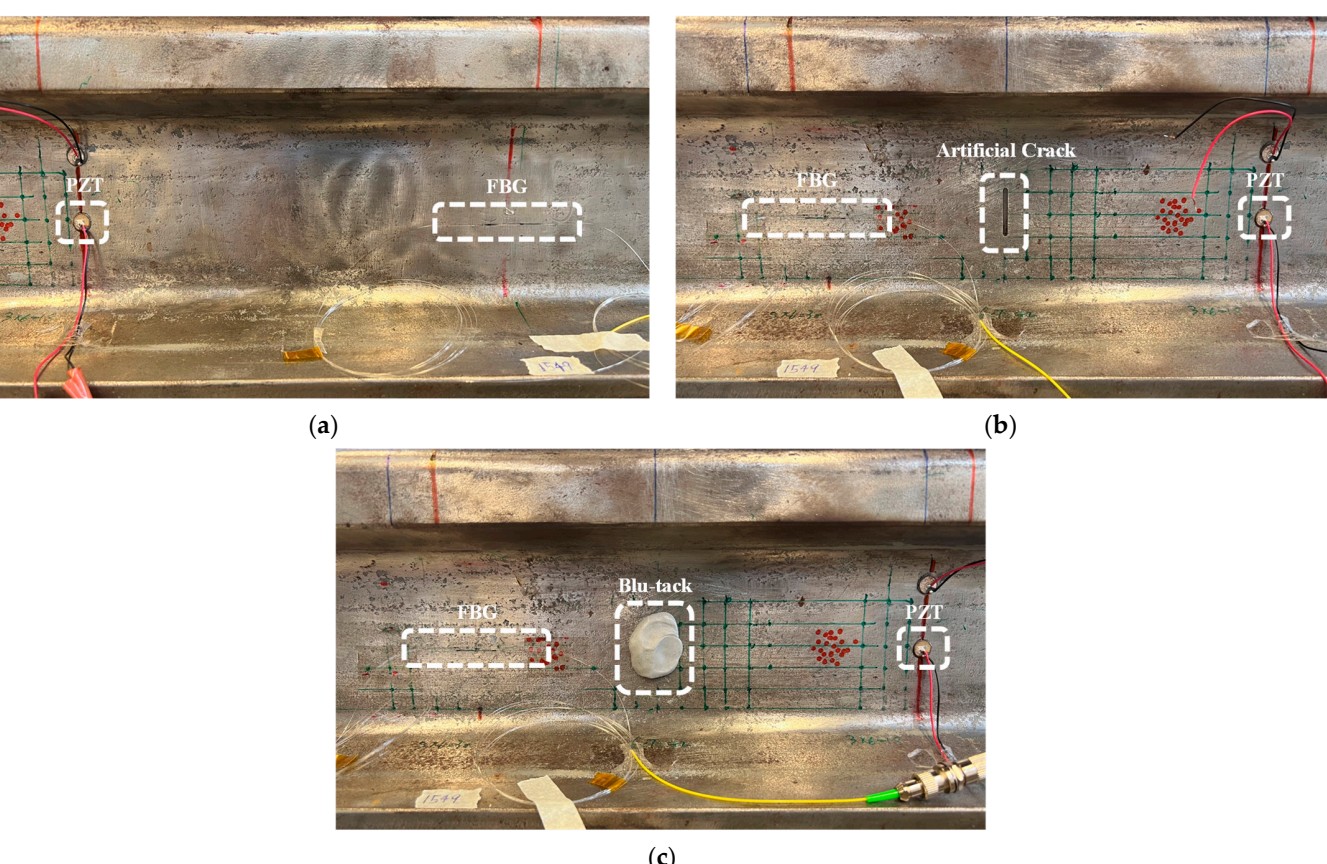

(**a**)  (**b**)

(**c**)

**Figure 6.** The sensors arrangement for (**a**) the intact condition; (**b**) the damaged condition with an artificial crack; (**c**) the damaged condition covered with blu-tack adhesives.

An FBG and an AE sensor are, respectively, placed at each measuring spot. The gap between the sensors at the same measuring point is considerably minor so that the distance between the sensors (FBG and AE sensors at the same measuring point) and the PZT actuator is slightly different. For each condition, the test is repeated 500 times at different instants over several days with varying temperatures, ensuring the effect of discreteness of the datasets will not interfere with later analysis.

### 3.2. Results and Discussions

It is worth mentioning that in this research, the variation in arrival time has been neglected by integrating the arrival time so that the waveforms are aligned to compare their shapes. Significant attenuation in signal amplitude for both FBG and AE data, due to the damage incurred on the rail, is observed as shown in Figure 7. This observed attenuation is because the propagation of UGWs is interfered with causing some waves

to reflect or refract into the damaged area. For the FBG data, when a single artificial crack exists, the amplitude of the received signals is compromised by approximately 30%; when the area is covered by blu-tack adhesives the amplitude drops sharply till it reaches only half of that in the intact case, mainly because the large volume of blu-tack has absorbed much of the energy of the UGWs. Although FBG and AE data in the time domain have similar shapes to the waveforms, the amplitude decline can be less notable for the AE time-domain data. Moreover, it is noted that the first arrival waves for FBG data are more distinguishable compared to AE data. After the first arrival waves, the reflected waves can also be recognized. However, due to the complexity of the railway shape the sources of the reflective waves after the first arrival waves are untraceable. The noise level of AE sensors interfered the receiving quality. Thus, it is fair to conclude that FBGs have higher sensitivity in the time domain of detecting the existence of damage, while blu-tack adhesives and cracks have a similar effect on the propagation of the UGWs. The normalized signals shown in Figure 8 are obtained by FBGs, demonstrating the variation in time-domain characteristics caused by the crack and the bump on the railway surface. Due to the existence of the damage, the first arriving wave packet has shown obvious delay; as for the later arriving packet possibly because of the reflections, a significant difference was observed when compared to each other. It can also be seen that the effects caused by cracks and bumps are quite similar, both resulting in the guided wave arrival delay and amplitude decay.

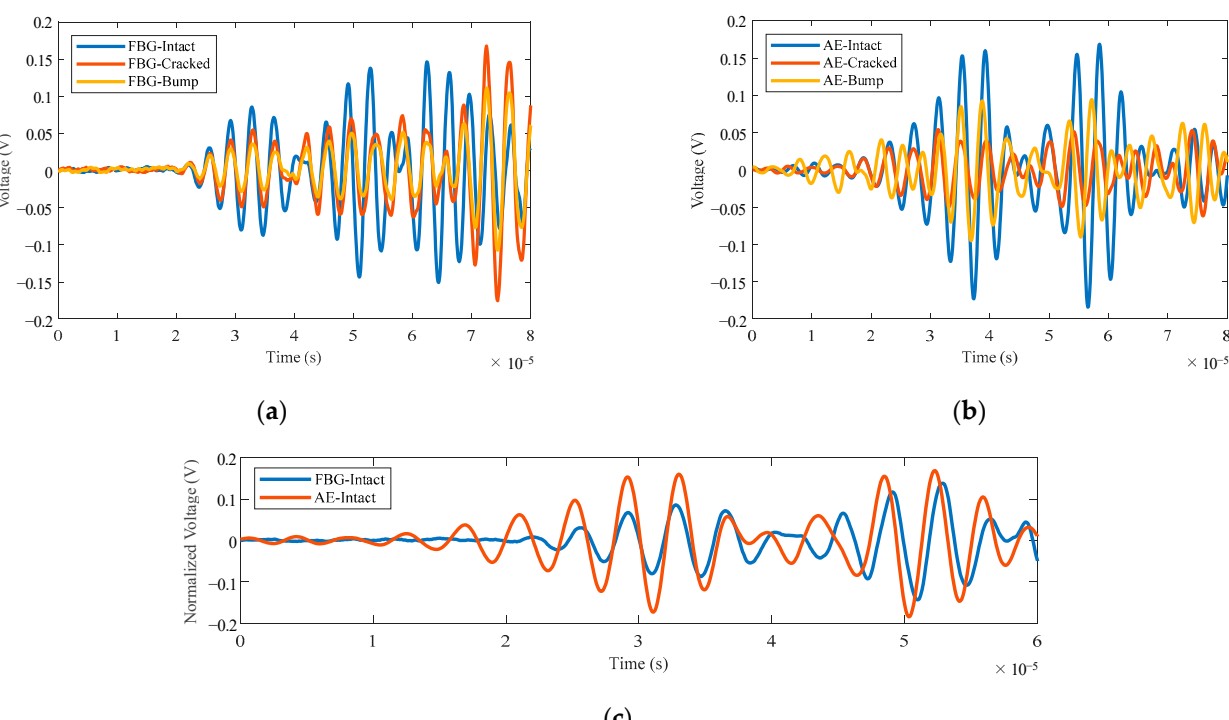

**Figure 7.** The time-domain signals under different conditions acquired by: (**a**) FBG sensors and (**b**) AE sensors; (**c**) Waveform comparison between FBG and AE signals under the intact condition.

The spectrums of both the FBG and AE signals are obtained through Fast Fourier Transform (FFT). Only slightly visible variations have been observed in the spectrums obtained from the experiments (Figure 9). The central frequency is approximately 250 kHz, which indicates that the UGW receiving quality is eligible. Due to the presence of the damage, the oscillation amplitude of the spectral amplitude shows a visible decline. However, the artificial crack and the blu-tack adhesives have dispersed the central frequencies, resulting in wider bandwidths of frequencies. Additionally, the noise level of FBG data is slightly higher than that of AE data, mainly caused by the fluctuations of the emitting laser wavelength which is mostly inevitable due to the limitation of the instrument precision.



Nevertheless, some low-frequency noise would not severely disrupt the subsequent analysis in this case. The spectrum amplitude has seen a slight drop in 'Cracked' and 'Bump' conditions, mainly because the waveform energy has reduced because of the defects on the propagation route.

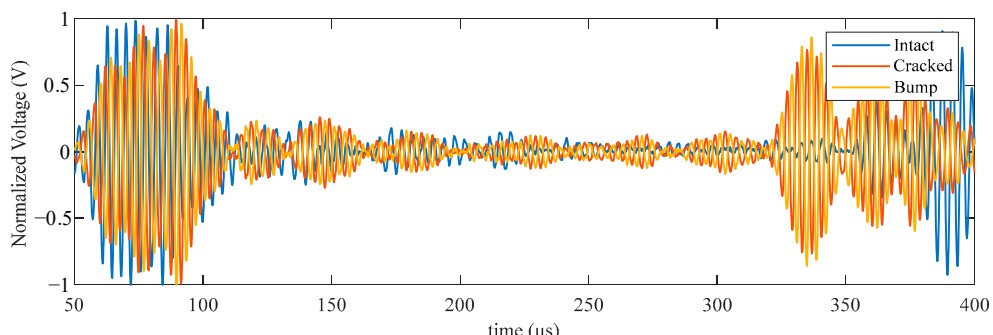

**Figure 8.** The UGW signals obtained by FBGs under different rail conditions.

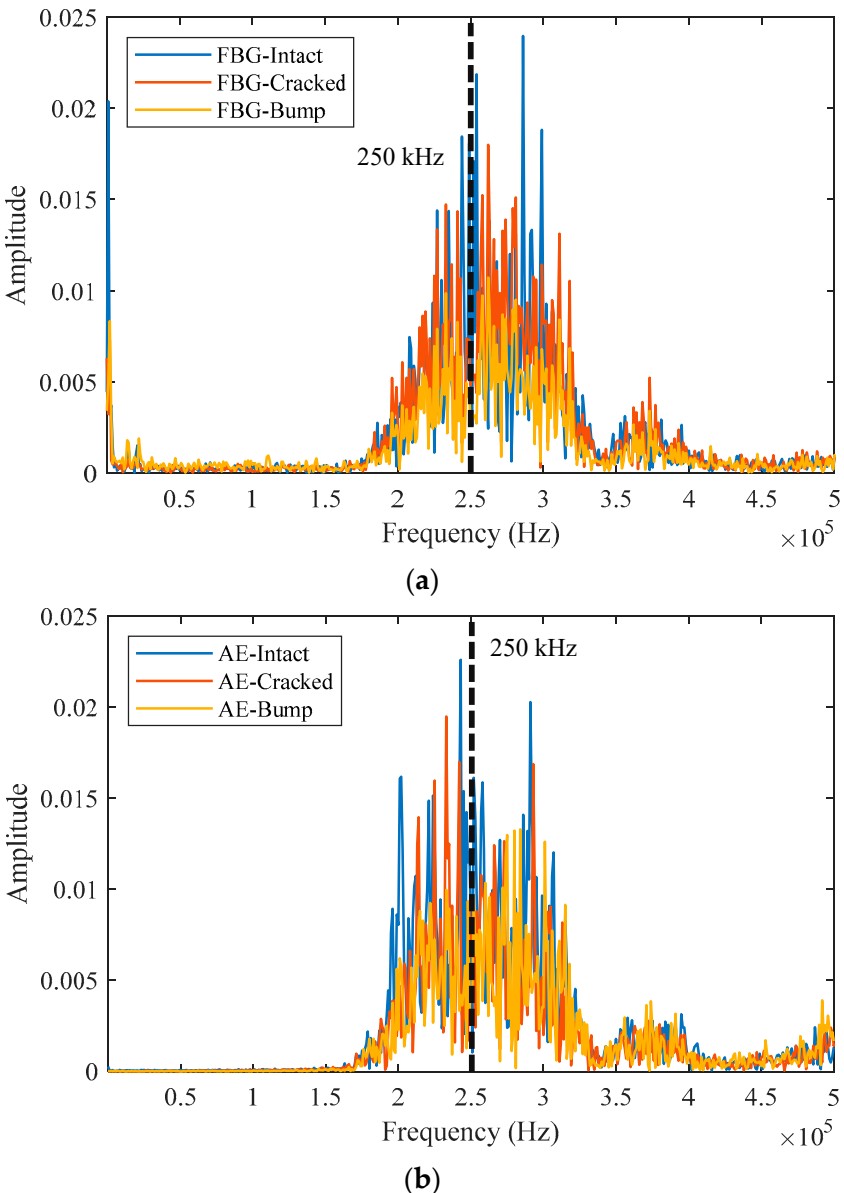

**Figure 9.** The frequency-domain plotting of the experimental data: (**a**) FBG data; (**b**) AE data.

Through the analysis in this section, it can be concluded that the defects on the rail will affect the UGW signals in time and frequency domains. With such observations, various damage indices have been proposed, such as waveform-distortion-based and energy-based indices [1]. The following section of this paper is devoted to the development and verification of a novel framework for rail defect detection using image classification to extract features from the UGW signals.

## 4. Image Classification-Based Rail Defect Detection

Although it has been discovered that damage on the rail will affect the propagation of UGWs in time and frequency domains, an effective and automatic diagnosis algorithm is still expected to be developed, reducing the labor cost of manual inspection and improving the identification accuracy.

### 4.1. Spectrogram Image Dataset

As illustrated in the previous section, significant changes caused by rail defects are observed. Based on the experimental data, the spectrograms are calculated using Short-time Fourier Transform (STFT), which contain characteristics in both time and frequency domains. The STFT is generally used to demonstrate the sinusoidal frequency and phase content as the signals vary over time, by dividing the raw time series into shorter segments. The calculation for a discrete-time STFT, represented by function F, can be expressed by:

$$\text{F}\{x[n]\}(m, \omega) = \sum_{n=-\infty}^{\infty} x[n]w(n-m)e^{-j\omega m} \tag{4}$$

where $x[n]$ is the discrete signal in time domain; $m$ is the number of points in the equal time gap of each short-time segment; $\omega$ is the frequency; $w(i)$ is the window function (commonly a Hann window or a Gaussian window) for some piece of series in the time domain. However, the commonly used calculation strategy in signal processing is to manually set a fixed time gap and apply the Fast Fourier transform (FFT) to each individual short time series, which is more efficient.

Three-dimension spectrograms have been obtained by STFT and chosen to represent the features for each test. As shown in Figure 10, the y- and z-axes are the frequency value and the amplitude of the spectrum, respectively, with time being the independent variable in this case. This figure shows the ultrasound signals received by an FBG at a single excitation. The central frequency lies at approximately 250 kHz, with the amplitude fluctuating through time. Although this three-dimension surface can represent the distinguishing features, it is not an ideal choice of input format for a CNN because the figure dimension is too high due to its redundancy and complexity.

However, it is applicable if the amplitude values are represented by different degrees of a color map (Figure 11a). The determination of the parameters of the STFT depends on the desired input image size to the CNN. In this case, the image size of $(32 \times 32 \times 3)$ is priorly confirmed, as a reference to the data pre-processing process. First, the signals with the length of 10,000 are divided into sections of length 256, windowed with a Hann window, having 200 samples overlapping between adjoining sections. Then, for each FFT sub-spectrum the frequency range is 10 kHz and the total time bins are 174. However, going through this process would end up generating images of the size of $(50 \times 174 \times 3)$, which is clearly redundant. Only the pixels that contain useful information are needed and extracted to form the images, as shown in Figure 11b. By controlling the time gap for each discrete color block, the spectrogram can be drawn in the 2-dimensional format. Extracting the entire spectrogram for the whole duration of each test would be inaccurate and unnecessary, hence the signals of the first arrival wave packet are utilized. Last, by eliminating the extra parts of the spectrograms, such as the label and the axis, the reflective images of the spectrograms with the resolution of 32 multiplied 32 pixels are obtained, each of which is an RGB image consisting of 3 element colors (red, green, and blue). The format shape for the process images is $(32 \times 32) \times 3$, which can be fed directly to the input layer of a CNN as shown in Figure 11b.

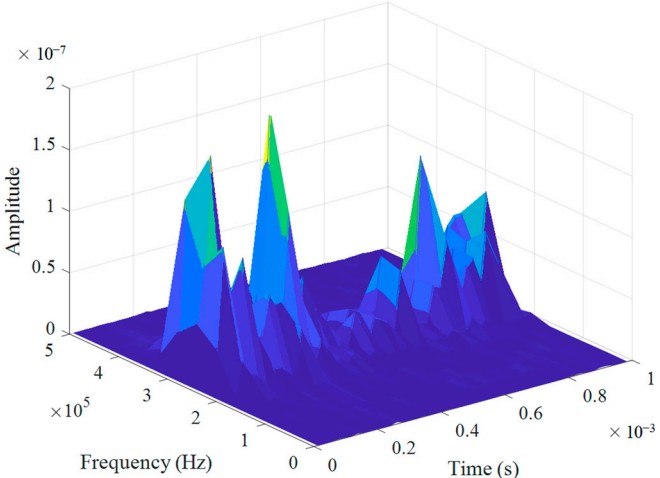

**Figure 10.** The 3-D spectrogram of the UGW data acquired by FBG.

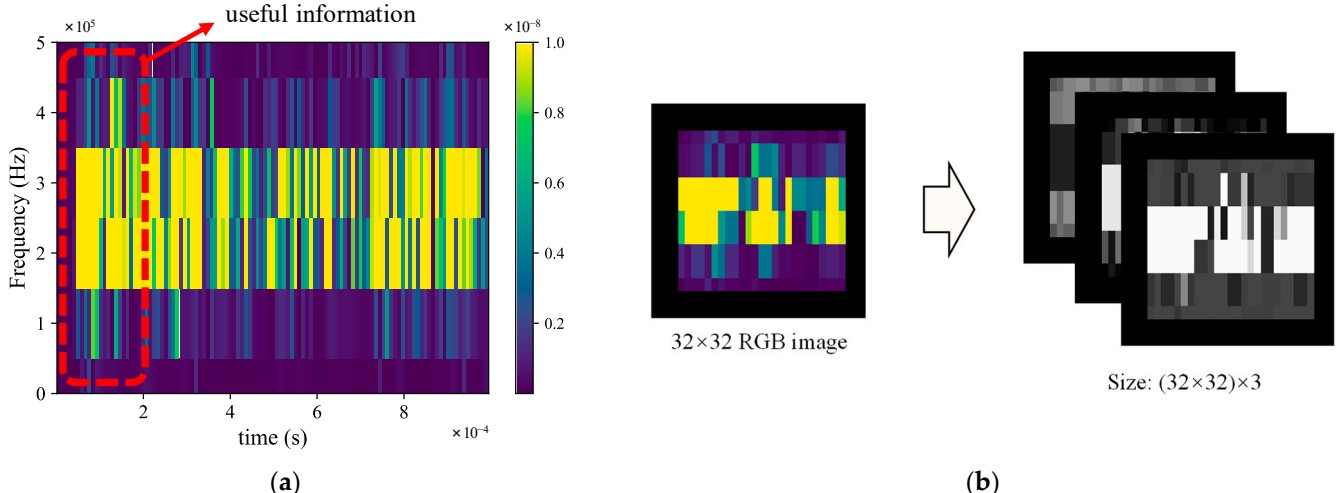

**Figure 11.** The imaging data pre-processing: (**a**) 2-D spectrogram with a color bar; (**b**) the extracted RGB image as the input of CNN.

*4.2. CNN*

CNNs have been widely utilized in the field of SHM regarding vision-based approaches [27,33,34]. Such machine learning algorithms have the advances in processing image datasets and achieving high accuracy in multi-class tasks. The general input for such networks is in the format of a tensor with the shape of (height) × (width) × (number of inputs) × (input channels). By arranging the CNN layers for specific circumstances, better performance on the validation set can be obtained.

The layer structure of the proposed CNN and the names for each layer in this study are shown in Figure 12. A total of three convolution layers, two max-pooling layers and one average-pooling layer have constructed the main computation procedures. The convolution layers function as the feature extraction processers. Specifically, they extract the useful information from an image and translate it into a feature map. The convolution computation is defined as an integration of two functions with one of them being reversed and shifted. For the convolution of 2-D images, it can be comprehended that the target image, which is the input of this layer, is convolutionally calculated with a specifically chosen kernel function. Then a new image is generated which is called a feature map. For this case with RGB images, 3 convolutional kernels are generated for each convolutional layer to process the inputs.

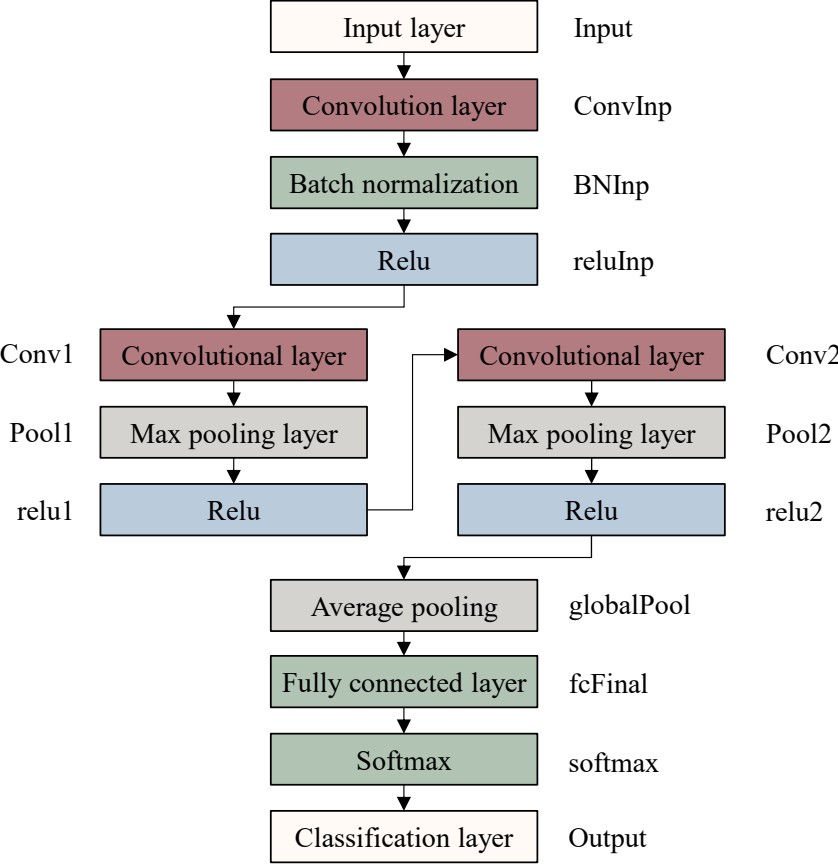

**Figure 12.** The CNN layer graph.

The pooling process is set to reduce the dimensions of data by using a tiling unit, which combines the outputs of the previous layer into clusters. This operation is extremely significant for preventing the network from over-fitting and gradient-disappearing. Lastly, a fully connected layer and a SoftMax layer process the output of the results transmitted from the above layers and reshape the 3D outputs into 1D outputs, as commonly seen in regular neural networks. The classification layer outputs the results, which in this case is the health condition: 'Intact', 'Cracked', and 'Bump', which have been encoded into float digits '0', '1', and '2'.

The specific parameters for each layer are listed in Table 2 below. To avoid the gradient-disappearing and over-fitting problems, the depth of the network has been strictly reduced considering that the size of the input data is limited.

**Table 2.** The layer settings of the proposed CNN.

| Network Layer | Layer Name | Parameters |
| --- | --- | --- |
| Input layer | 'Input' | Input image size = (32, 32, 3) |
| 2-D Convolution layer | 'ConvInp' | net width = 16, filter size = 3, padding mode = 'same' |
| Batch normalization layer | 'BNInp' | minimum batch number = 16 |
| Relu layer | 'reluInp' | Relu activation function |
| 2-D Convolution layer | 'Conv1' | net width = 16, filter size = 1, padding mode = 'same' |
| 2-D Max pooling layer | 'Pool1' | pool size = 2 |
| Relu layer | 'relu1' | Relu activation function |
| 2-D Convolution layer | 'Conv2' | net width = 16, filter size = 1, padding mode = 'same' |
| 2-D Max pooling layer | 'Pool2' | pool size = 2 |
| Relu layer | 'relu2' | Relu activation function |
| Average pooling layer | 'globalPool' | pool size = 8 |
| Fully connected layer | 'fcFinal' | output size = 3 |
| Softmax layer | 'softmax' | Softmax activation function |
| Classification layer | 'Output' | number of classes = 3 |

### 4.3. Classification Results

The training process is completed via MATLAB R2022a. The minimum batch size for the training is 16 images, with the validation frequency set to be every 30 rounds of updating. The learning rate is initially 0.01, with a fixed dropping rate of 10% for each epoch. Considering the number of images for this classification is not large, the stochastic gradient descent with momentum (SGDM) optimizer is utilized to update the parameters for each layer. The stopping criterion for the training is the validation accuracy staying the same for 4 epochs. The training set is composed of a total of 900 images, with 300 images for each class, respectively. As for the validation, 300 images are randomly selected with 100 images for each class ('Intact', 'Cracked' and 'Bump'). For this experiment, the training stopped at iteration 336 (Figure 13a).

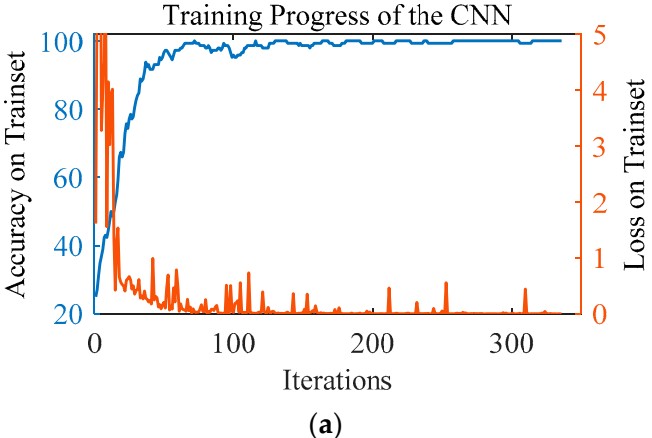

**Figure 13.** Classification results of the CNN: (**a**) The training accuracy and loss on trainset; (**b**) The confusion matrix of the validation results.

From the training record shown in Figure 13b it can be concluded that the network performs well on the trainset, with a training error of only 0.1%. In contrast, the validation error is slightly higher, but still admissible, at 2.0%. To further demonstrate the classification results, classic measuring indices in the statistical analysis of multi-class machine learning tasks are introduced:

$$
\begin{aligned}
precision &= \frac{tp}{tp+fp} \\
recall &= \frac{tp}{tp+fn} \\
accuracy &= \frac{tp+tn}{tp+tn+fp+fn} \\
F_1 &= 2\frac{precision \cdot recall}{precision+recall}
\end{aligned}
\tag{5}
$$

where $tp$, $tn$, $fp$ and $fn$ are shorts for 'true positive', 'true negative', 'false positive' and 'false negative' predictions of a classifier.

The drawn confusion matrix indicates that only a few samples in the validation set are misclassified. The precision and recall values are more than 97.0% and the overall accuracy is 98.0%. Additionally, the convergence speed of the network is also exceptionally fast, proving that the proposed approach is highly efficient and applicable. The $F_1$ score in this experiment for each class is 100.0%, 97.0%, 97.0%. It is also worth mentioning that the CNN has 100.0% accuracy in identifying the damaged conditions from the healthy ones, which is of great significance in real engineering scenarios.

### 4.4. Visualization of the CNN

The feature images shown in Figure 14 are extracted from the hidden layers to analyze the performance of the framework. A total of 16 feature images have been obtained for each convolution layer. It is assumed that the more distinguished the image is, the better the classification results. Here, one individual channel has been selected for each convolution

layer to show visualized output images, as circled in Figure 14. Then, based on the channel selected, the visualization of the classification has been shown in Figure 15, taking three images from the validation set as an example to illustrate the image processing. It can be observed that although the 'ConvInp' layer can barely capture differential features of the three input images, the outputs for 'Conv1' and 'Conv2' layers succeed in extracting these homogeneous photos into three clear categories.

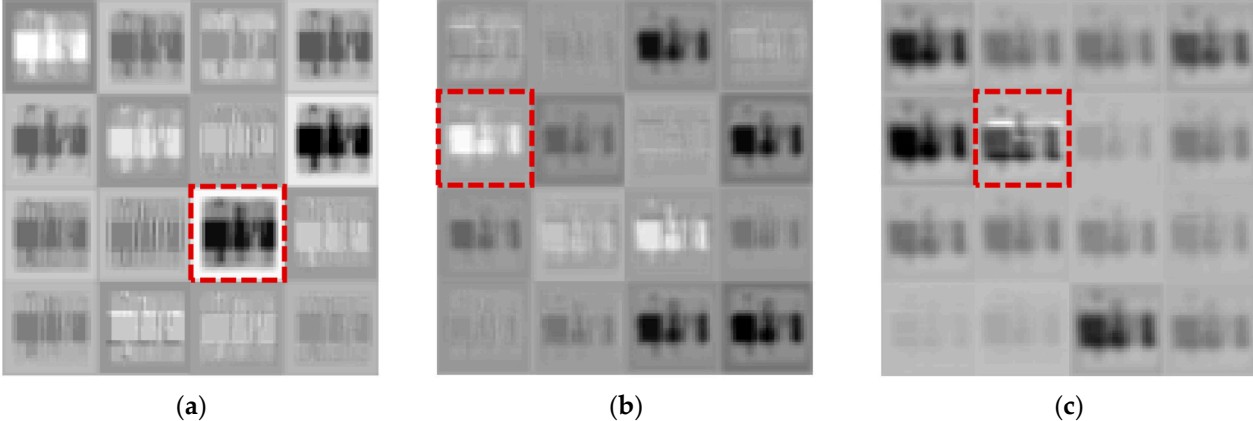

(**a**)  (**b**)  (**c**)

**Figure 14.** The 16 feature images for each convolution layer: (**a**) ConvInp; (**b**) Conv1; (**c**) Conv2.

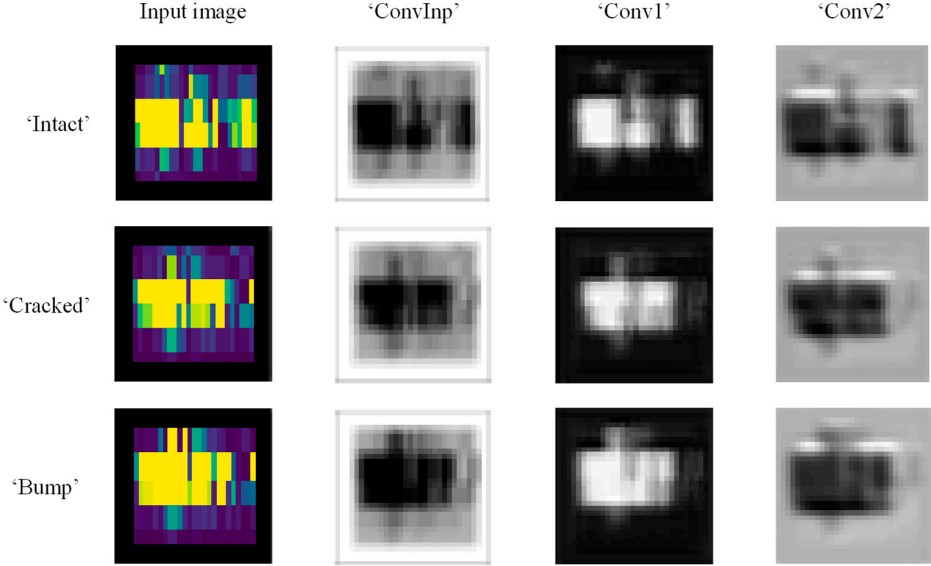

**Figure 15.** Visualization of the feature extraction results of each convolutional layer for each class.

Figure 16 shows part of the final classification results with the corresponding raw RGB images and the prediction probabilities. Although it can be observed that the images from where the rail has a crack are similar to those with blu-tack adhesives, the designed CNN still manages to identify most of the images. That undoubtedly proves the robustness of this machine learning algorithm in performing multi-classification tasks.

Additionally, to further reveal the classification principals of all three sub-channels developed for the image input of each color value, the 16 feature maps of one of the 2-D convolution layers, namely 'Conv1', are demonstrated in Figure 17. Each map has been split into 3 sub-maps according to its RGB values. It can be observed that the training outcomes for each input single-color image are quite nonidentical, yet the feature map of which RGB color makes more contributions to the accurate classification results cannot be obtained prior to the training process. Therefore, it is evident that RBG images, compared with 2-D grayscale images, generally contain more information that could be extracted and learnt by the network, although the former requires more computational resource.

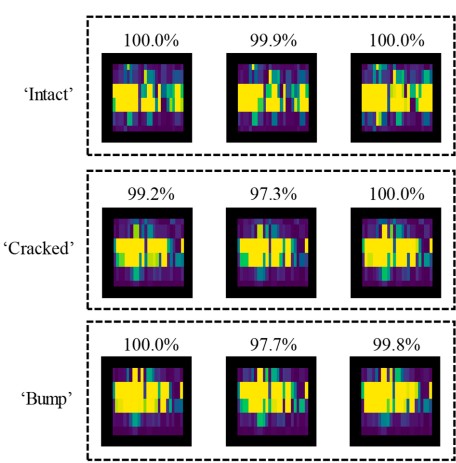

**Figure 16.** Visualization of the classification results.

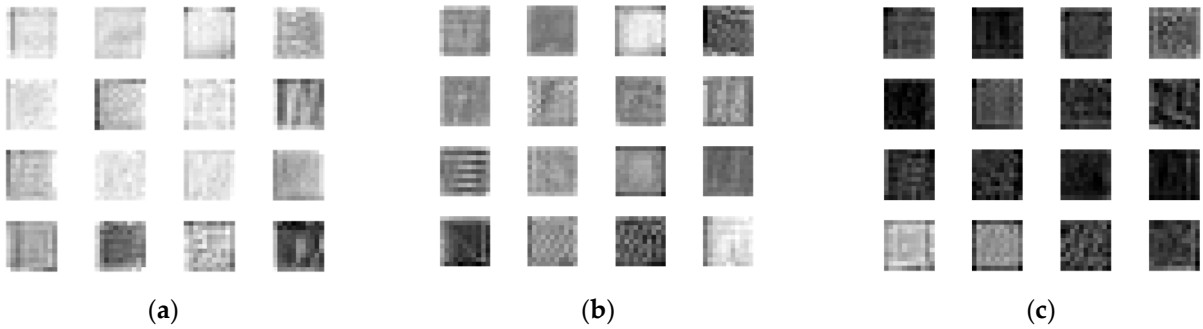

(a)                                        (b)                                        (c)

**Figure 17.** The feature maps of 3 channels generated by the 'Conv1' layer: (**a**) 'R' channel; (**b**) 'G' channel; (**c**) 'B' channel.

## 5. Conclusions

In this paper, a novel image classification approach has been proposed to identify the rail track health status using UGW inspection. First, a hybrid sensing system composed of a PZT actuator and ultrasound FBG sensors has been proposed to conduct experiments. An artificial crack and a surface bump made of blu-tack adhesives are considered two damaged cases in the experiments, respectively. Through comparative studies between the UGW signals acquired by FBG and AE sensors, significant changes in waveforms in time and frequency domains have been observed. It has been concluded that FBGs are consummate sensors to be massively utilized in UGW inspection tasks, especially in complex environments with high-level EMI. A further contribution of this paper focuses on proposing a specially designed CNN-based approach to efficiently identify the health conditions of the rail tracks. Experimental ultrasound data under intact and defective conditions is pre-processed using STFT considering that the frequency proportion of the wave packets varies with the propagation period. The obtained spectrograms are then utilized to feed the proposed CNN model. The results have been visualized for comparison, showing high accuracy for the prediction of each rail condition. After 336 iterations, the proposed classifier achieves 98.0% overall classification accuracy with a fast convergence speed. The visualization of each convolution layer of the classifier has also been shown to illustrate the feature extraction process of the CNN.

Consequently, the methodology proposed in this paper has the potential to be massively utilized for on-site monitoring, considering that FBGs can avoid the system disfunction caused by EMI. The image classification-based damage detection has been verified in this paper and will continue to contribute to the automatic online damage detection of railway tracks to save manual labor with less time consumption. However, for future research to achieve damage location and even quantification, more FBGs are required to be

deployed on the rail track for more-precise measurements. It is applicable to predict the approximate damage location along the railway line that lasts for kilometers if adequate measuring points are distributed. As for the determination of damage severity, more advanced damage indices will be developed to achieve the quantification.

**Author Contributions:** Conceptualization, D.-Z.D. and Y.-Q.N.; methodology, D.-Z.D. and C.-C.L.; software, D.-Z.D.; validation, D.-Z.D., C.-C.L. and Q.Z.; formal analysis, D.-Z.D. and C.-C.L.; investigation, Q.Z. and B.S.; resources, Y.-Q.N.; data curation, D.-Z.D., B.S. and Q.-F.Z.; writing—original draft preparation, D.-Z.D. and C.-C.L.; writing—review and editing, Y.-Q.N. and Q.Z.; visualization, Q.-F.Z.; supervision, Y.-Q.N.; project administration, Y.-Q.N.; funding acquisition, Y.-Q.N. All authors have read and agreed to the published version of the manuscript.

**Funding:** This research was funded by a grant from the Research Grants Council of the Hong Kong Special Administrative Region (SAR), China (Grant Number: R-5020-18) and a grant from the Science, Technology and Innovation Commission of Shenzhen Municipality under Central-Guided Local Technology Development Fund (Grant Number: 2021SZVUP143). The authors would also like to appreciate the funding support by the Innovation and Technology Commission of the Hong Kong SAR Government to the Hong Kong Branch of National Rail Transit Electrification and Automation Engineering Technology Research Center (Grant Number: K-BBY1).

**Institutional Review Board Statement:** Not applicable.

**Informed Consent Statement:** Not applicable.

**Data Availability Statement:** Not applicable.

**Conflicts of Interest:** The authors declare no conflict of interest.

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
