# Peer review of "Image Classification-Based Defect Detection of Railway Tracks Using Fiber Bragg Grating Ultrasonic Sensors"

_applsci, doi:10.3390/app13010384_

Round 1

Reviewer 1 Report

The manuscript entitled “Image Classification-based Defect Detection of Railway Tracks Using Fiber Bragg Grating Ultrasonic Sensors” proposes a method for detecting damage to railway tracks based on CNN classification of input images (spectrograms) acquired by the STFT of UGW signals obtained by FBG sensors. The method shows promising results in the proposed practical application.

Here are some comments I would like the authors to address before the manuscript is considered for publication:

1.      Please add “convolutional neural network (CNN)” and “Short-time Fourier Transform (STFT)”  as additional keywords, as they are essential parts of the study. This will allow an interested reader to find the paper more easily.

2.      Please provide a brief overview of the manuscript’s structure in the last paragraph of the Introduction section.

3.      The literature review and related work section are well written, addressing relevant recent studies, correctly placing the manuscript within the research field, and stating the motivation for this study oriented towards a specific practical application in railways. However, the application of deep CNNs with various 2D time-frequency signal representations, including spectrograms, has become a hot research topic recently. Therefore, I would like to suggest the authors supplement the introductory part with some of the recent studies on this topic to briefly illustrate the state-of-the-art performances of the CNNs and time-frequency representations in many different applications today and provide an interested reader with examples. Please consider briefly mentioning the following papers for illustration purposes: 10.1007/s10044-020-00921-5, 10.1109/ACCESS.2021.3139850, 10.1109/TNNLS.2020.3008938.

4.      Please elaborate more on the selected parameters of the STFT’s sliding window. Is this selection optimized?

5.      Please provide arguments for using RGB images of spectrograms (three 2D 32x32 arrays) as input to the CNN. Would a grayscale (intensity) spectrogram image not be sufficient, as it would take less memory?

6.      Did the authors consider using any techniques for data augmentation, as the dataset size is limited?

7.      Please provide the training parameters for the CNN.

8.      In the Conclusion section, please address the limitations of the proposed method.

9.      In the Conclusion section, please also provide some directions for future research.

Reviewer 2 Report

The author developed a piezoelectric/optical fiber hybrid sensing system to collect ultrasonic guided wave signals, and proposed an image classification method to classify the damage in railway track. The paper is well written, but there are the following problems:

1)Piezoelectric/optical fiber hybrid guided wave sensing system has been available for a long time. Where is the innovation of the system you developed?

2)The author has not fully investigated the current damage monitoring methods based on deep learning and ultrasonic guided waves, and some new studies have not been mentioned.

3)Why can cracks and bump damage be classified? Can we see the difference from some characteristics of the physical signals caused by the two kinds of damage?

4)The method proposed by the author only gives a three classification, is it too simple? Why are there only two kinds of injuries? Are there only two types of damage on the railway track?

5)Can the authors predict the performance of the proposed method for damage location and quantification?

Round 2

Reviewer 1 Report

The authors have addressed my comments and significantly improved the manuscript's quality.

Reviewer 2 Report

The authors have addressed all of my questions. The manuscript can be published in the current form.